# SoMA: A Real-to-Sim Neural Simulator for Robotic Soft-body Manipulation

Mu Huang [1 2]   Hui Wang [3 2]   Kerui Ren [3 2]   Linning Xu [4 2]   Yunsong Zhou [2]   Mulin Yu [2]
Bo Dai[† 5] Jiangmiao Pang [2]

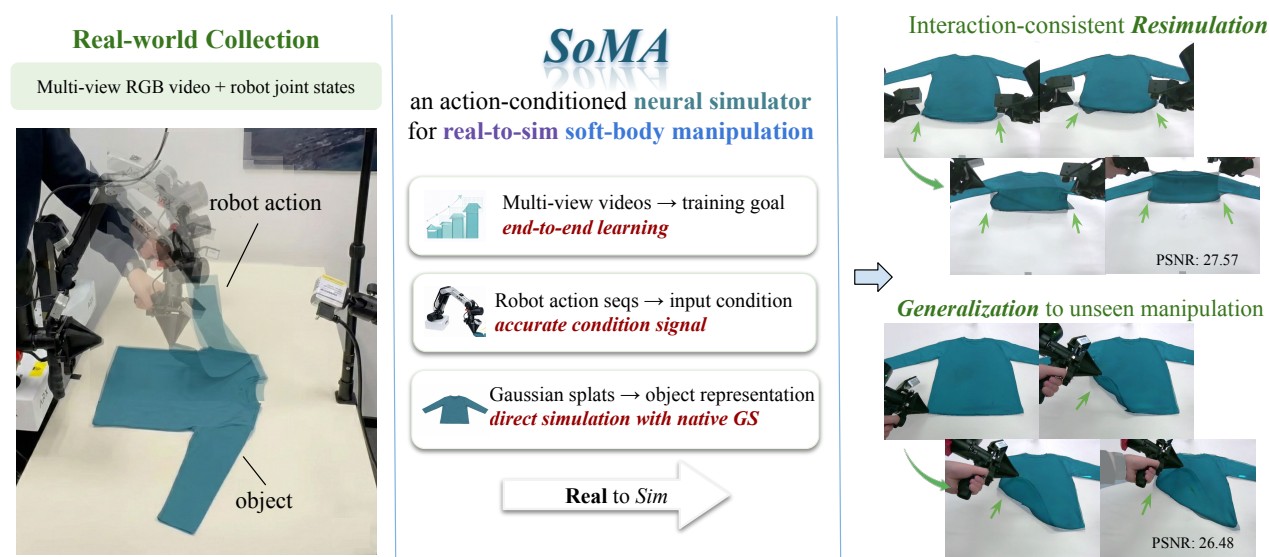

*Figure 1.* **SoMA** is a GS neural simulator that *reconstructs* and *simulates* deformable object dynamics from real-world robot manipulation. Learning from multi-view RGB observations, it performs action-conditioned simulation directly on Gaussian splats, enabling interaction-consistent, stable long-horizon resimulation with higher-fidelity rendering under both seen and unseen manipulations.

## Abstract

Simulating deformable objects under rich interactions remains a fundamental challenge for real-to-sim robot manipulation, with dynamics jointly driven by environmental effects and robot actions. Existing simulators rely on predefined physics or data-driven dynamics without robot-conditioned control, limiting accuracy, stability, and generalization. This paper presents **SoMA**, a 3D Gaussian Splat simulator for soft-body manipulation. SoMA couples deformable dynamics, environmental forces, and robot joint actions in a unified latent neural space for end-to-end real-to-sim simulation. Modeling interactions over learned Gaussian splats enables controllable, stable long-horizon manipulation and generalization beyond observed trajectories without predefined physical models. SoMA improves resimulation accuracy and generalization on real-world robot manipulation by 20%, enabling stable simulation of complex tasks such as long-horizon cloth folding. *Project Page*: city-super.github.io/SoMA

## 1. Introduction

Embodied learning is inherently data-driven, as it requires large-scale interaction data for robots to understand and act in the physical world. Yet, collecting real-world robot manipulation data is costly and risky (Zheng et al., 2025), as in tasks involving deformable objects like cloth folding and flexible object handling (Ganapathi et al., 2021; Mo et al., 2022). Thus, real-to-sim (R2S) simulation offers a alternative by scalable reproducing real-world object behaviors in virtual environments, serving as a foundation for data synthesis, augmentation, and policy learning.

A practical simulator must strike a balance between *physical fidelity* and *long-horizon interaction consistency*. It should

[1]Fudan University, China [2]Shanghai Artificial Intelligence Laboratory, China [3]Shanghai Jiao Tong University, China [4]The Chinese University of Hong Kong, China [5]The University of Hong Kong, China. Correspondence to: Bo Dai <bdai@hku.hk>.

*Proceedings of the $43^{rd}$ International Conference on Machine Learning*, Seoul, South Korea. PMLR 306, 2026. Copyright 2026 by the author(s).

faithfully capture deformable object geometry and dynamics without introducing R2S bias, while remaining stable and coherent under sustained robot–object interaction. While geometry can often be recovered from visual observations, learning physically meaningful dynamics and interactions remains challenging. This challenge is exacerbated by partial occlusion and complex contact patterns.

Existing approaches address these challenges along two largely separate directions. Physics-based simulators (Belytschko et al., 2014; Jiang et al., 2016) ensure consistent interaction over long horizons, but rely on predefined physical models and parameters that are difficult to infer from visual data. Differentiable simulators (Jiang et al., 2025a; Zhang et al., 2024b; Zhong et al., 2024) alleviate this limitation by optimizing a small set of parameters, yet remain constrained by simplified physical assumptions. In contrast, neural dynamics modeling (Shao et al., 2025; Zhang et al., 2024a) and 4D reconstruction methods (Singer et al., 2023a; Bahmani et al., 2025; Zhang et al., 2025; Ren et al., 2023) learn motion directly from data and recover dynamic geometry, but primarily focus on reproducing observed trajectories, offering limited support for robot-conditioned interaction and generalization beyond the training distribution. As a result, neither direction alone suffices for R2S simulation in complex robot manipulation.

We bridge the two worlds by rethinking the representation and learning of deformable object simulation. This paper proposes **SoMA**, a real-to-sim neural Simulator of Manipulation for soft-body interaction, operating directly on learned Gaussian splat representations. SoMA models deformable objects, robot actions, and environmental effects in a unified learned particle representation, enabling causal action-driven dynamics and stable long-horizon interaction without predefined physical rules.

Explicit designs are proposed to achieve the functionality through the neural simulator. To ensure kinematically consistent interaction between the robot and deformable objects, it establishes a robot-conditioned R2S mapping that anchors learned object dynamics to the robot's joint-space actions, ensuring causal and kinematically consistent interaction. For physically consistent interaction, dynamics are modeled through force-driven updates defined directly on Gaussian splats, allowing local contact effects to propagate through the deformable object even when visual observations are partial or occluded. To support long-horizon interaction without drift or collapse, SoMA adopts a multi-resolution training strategy that balances temporal coverage and computational efficiency, together with a blended supervision scheme that combines occlusion-aware reconstruction signals with physics-inspired consistency constraints. Together, these components enable SoMA to serve as a stable, action-conditioned neural simulator for Real-to-Sim manipulation.

We summarize our contributions as follows: (1) A R2S neural simulation paradigm for soft-body manipulation, supporting long-horizon, interaction-consistent simulation. (2) Mechanisms that make such simulation feasible, including a robot-conditioned alignment to anchor scenes to real kinematics, a force-driven Gaussian-splat dynamics model for contact and occlusion, and multi-resolution training with blended supervision for stable long-horizon performance. (3) Extensive evaluation on public benchmarks and a new real-world dataset, demonstrating state-of-the-art RGB and depth performance (20% improvement) and clear advantages for sustained interactive manipulation.

## 2. Related Works

**3D reconstruction and scene representation.** Camera pose estimation and 3D reconstruction form the geometric basis of vision-based simulation and real-to-sim pipelines. Classical multi-view geometry methods, such as SLAM and COLMAP (Cadena et al., 2017; Campos et al., 2021; Grisetti et al., 2011; Schonberger & Frahm, 2016), recover camera poses and scene structure from overlapping views. Recent feed-forward reconstruction approaches based on Gaussian Splatting, including VGGT, Pi3, and AnySplat (Wang et al., 2025b; Jiang et al., 2025b; Kerbl et al., 2023; Liu et al., 2025; Wang et al., 2025a), further enable reconstruction under sparse or unstructured observations. Our work builds on these methods to obtain camera poses and Gaussian splat representations, while focusing on dynamics modeling and robot-conditioned interaction rather than reconstruction.

**Physics-based simulators.** Physics-based simulators, such as FEM, MPM, and SPH (Belytschko et al., 2014; Gingold & Monaghan, 1977; Jiang et al., 2016; Müller et al., 2007; Reddy, 1993; Sulsky et al., 1994), are widely used to model deformable objects under controlled settings and are supported by embodied simulation platforms such as Isaac Sim (NVIDIA, 2023). However, these methods rely on carefully specified physical parameters and simulator configurations, making accurate parameter identification from visual observations challenging in real-to-sim scenarios, especially under robot-driven interactions. Recent differentiable simulators (Hu et al.), including PhysDreamer and PhysTwin (Zhang et al., 2024b; Zhong et al., 2024; Jiang et al., 2025a), attempt inverse optimization of limited parameters but still depend on predefined physical models and simplified structures, limiting their ability to capture complex real-world motions and interactions.

**Neural-based dynamics modeling.** Neural-based approaches learn object dynamics directly from data, reducing reliance on explicit physical parameter specification. Some 4D reconstruction methods (Singer et al., 2023a; Bahmani et al., 2025; Borycki et al., 2026; Feng et al., 2024; Zhang

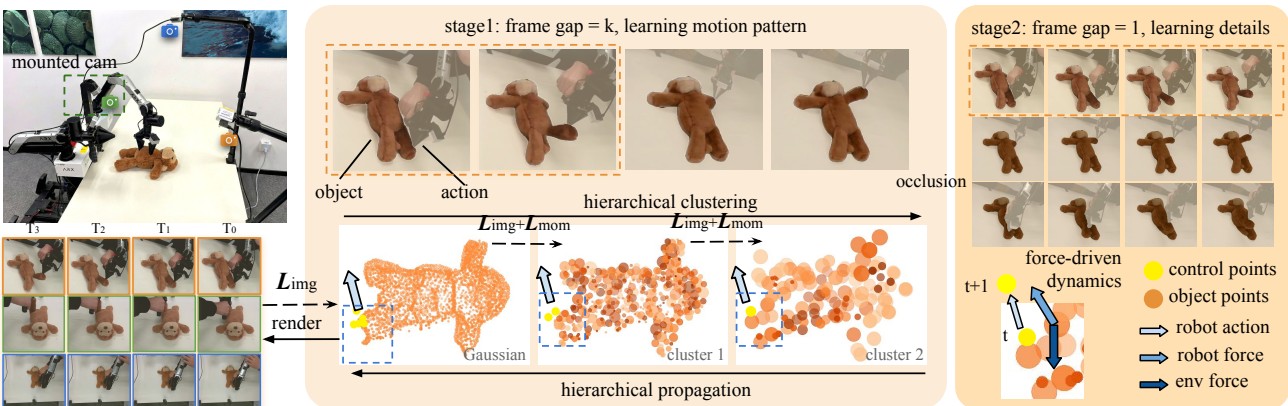

*Figure 2.* **Framework of SoMA.** SoMA takes RGB observations and robot joint-space actions collected from real-world manipulation as input (Left). It reconstructs deformable objects as hierarchical Gaussian splats, and propagates them through a neural simulator with supervision from rendering and dynamics (Middle). Object motion is driven by force-based interactions, where environmental and robot-induced forces act on splats to produce deformation (Right). A two-stage multi-resolution training strategy first captures global motion with large temporal gaps and then refines fine-grained dynamics under occlusion and contact using small gaps.

et al., 2025; Ren et al., 2023; Shen et al., 2023; Singer et al., 2023b; Yin et al., 2023) extend Gaussian Splatting to dynamic scenes, providing temporally consistent geometry but mainly reconstruct observed motions with limited interaction modeling. Neural simulators on GS representations, such as GausSim and GS-Dynamics (Shao et al., 2025; Zhang et al., 2024a), regress future states from past states, improving generalization over reconstruction-based methods but typically evaluated under simplified objects and interaction settings. In contrast, our method targets robot-manipulated, occlusion-heavy embodied scenarios by explicitly modeling robot–object interactions and performing end-to-end simulation directly from video observations.

## 3. Preliminaries

**Hierarchical graph simulator.** We represents deformable object as a collection of Gaussian splats (GS) (Shao et al., 2025; Kerbl et al., 2023), where each splat $g_i = \{\mathbf{x}_i, \mathbf{\Sigma}_i, m_i, \mathbf{a}_i\}$ encodes its spatial position $\mathbf{x}_i \in \mathbb{R}^3$, anisotropic covariance $\mathbf{\Sigma}_i \in \mathbb{R}^{3\times3}$, mass $m_i$, and additional physical attributes $\mathbf{a}_i \in \mathbb{R}^{\texttt{attr\_dim}}$. To improve efficiency and better capture the continuous dynamics of deformable objects, the GS are organized into a hierarchical graph structure. Starting from the finest GS level, splats are recursively clustered to form higher-level nodes, each representing an aggregated local structure of the object. For a cluster composed of $N_i$ child nodes, its aggregated attributes are computed by combining the physical states of its child nodes:

$$m_{cl} = \sum_i m_i, \quad \mathbf{x}_{cl} = \frac{\sum_i m_i \mathbf{x}_i}{m_{cl}}, \quad \mathbf{a}_{cl} = \frac{1}{N_i} \sum_i \mathbf{a}_i. \tag{1}$$

This bottom-up construction defines a multi-level hierarchical graph that captures object structure at different spatial scales. Dynamics are then propagated through the hierarchy using graph neural networks in a top-down manner. At each level, latent motion and deformation states are predicted for cluster nodes and subsequently transferred to their children through learned transformations. Following prior hierarchical GS simulators, the state of a GS node $k$ at level $h - 1$ is obtained from its ancestor cluster $c_h$ as:

$$\hat{\mathbf{x}}_k^{h-1} = \hat{\mathbf{x}}_{c_h}^h + \prod_{j=h}^{L} \mathbf{F}_k^j (\mathbf{X}_k - \mathbf{X}_{c_h}), \tag{2}$$

$$\hat{\mathbf{\Sigma}}_k^{h-1} = \left(\prod_{i=h}^{L} \mathbf{F}_k^i\right) \mathbf{\Sigma}_k \left(\prod_{i=h}^{L} \mathbf{F}_k^i\right)^{\top}, \tag{3}$$

where $\mathbf{F}_k^i$ denotes the learned transformation at level $i$. This hierarchical propagation enables coherent global motion while preserving local deformations at the GS level.

## 4. Method

### 4.1. Problem Definition

We consider modeling deformable object dynamics under robot manipulation from multi-view RGB videos. At each time step $t$, we are given synchronized RGB images $\mathbf{I}_{t,i}$ captured by three cameras, together with the robot joint state $\mathbf{R}_t$. An initial object state is reconstructed as a set of Gaussian splats $G_0 = \{g_i\}$. The object state is then regressed over time using an action-conditioned simulator:

$$G_t = \phi_\theta(G_{t-1}, G_{t-2}, \mathbf{R}_t), \tag{4}$$

where $\phi_\theta$ predicts the object dynamics conditioned on robot actions.

At each time step, the predicted state $G_t$ is rendered under known camera poses to obtain images $\hat{\mathbf{I}}_{t,i}$. The simulator is trained end-to-end by minimizing the reconstruction loss between rendered and observed images across time and views:

$$\min_{\theta,a} \sum_{t,i} \mathcal{L}(\hat{\mathbf{I}}_{t,i}, \mathbf{I}_{t,i}). \tag{5}$$

## 4.2. SoMA Framework

SoMA is a unified neural simulator for soft-body robot manipulation, designed to model deformable object dynamics under direct robot joint-space control and environmental interaction. Unlike prior approaches that decouple object dynamics from control signals or rely on externally specified end-effector trajectories, SoMA jointly represents robots, deformable objects, and environments within a shared simulation space. This unified formulation allows robot actions to directly drive object dynamics, enabling interaction-aware and numerically stable long-horizon simulation from visual observations.

As shown in Figure 2, SoMA consists of three key components. First, a scene-to-simulation mapping module lifts real-world observations into a unified simulation space, producing consistent representations of the robot, deformable object, and environment (§4.2.1). Second, a hierarchical graph-based simulator models robot–object–environment interactions through structured interaction graphs, enabling localized contact reasoning with global physical consistency (§4.2.2). Third, we adopt a multi-resolution training strategy to optimize dynamics across temporal and spatial scales for efficient and stable long-horizon simulation, together with a blended supervision scheme that combines occlusion-aware image losses and physics-inspired consistency constraints (§4.2.3, §4.2.4). Together, these components form an interaction-aware simulation pipeline for physically consistent soft-body manipulation.

### 4.2.1. SCENE INITIALIZATION VIA R2S MAPPING

Embodied manipulation poses a fundamental real-to-sim challenge: visual reconstructions, robot kinematics, and physical reference frames exist in heterogeneous coordinate systems and metric scales. Common initialization strategies that reconstruct objects alone or align object motion with tracked end-effector trajectories decouple geometry from control, overlooking robot kinematics and environmental constraints. As a result, joint-space actions cannot be directly applied, and physically meaningful contact reasoning becomes ill-defined. Therefore, we construct a unified simulation space consistent with robot kinematics, manipulated objects, and physical reference frames.

**Reconstruction.** Given multi-view RGB images from calibrated cameras, camera poses are estimated using multi-view geometry. Based on the recovered poses, the manipulated object are reconstructed using Gaussian Splatting (Kerbl et al., 2023), yielding a GS representation $G_0$ in the camera coordinate system.

**Robot Conditioning.** We recover the global scale factor $s$ by enforcing metric consistency between reconstructed geometry and known reference dimensions observed across the robot, camera, and world coordinate systems. The rigid transformation parameters $\mathbf{R}$ and $\mathbf{t}$ are estimated from the relative poses of the camera expressed in the robot and camera coordinates. Given the robot joint configuration $\mathbf{q}_t$ at time $t$, the end-effector pose in the robot base frame is obtained via forward kinematics:

$$\mathbf{T}^{ee}_{rob}(t) = \mathrm{FK}(\mathbf{q}_t). \tag{6}$$

The corresponding pose in the simulation space is computed by a similarity transformation:

$$\mathbf{T}^{ee}_{sim}(t) = \begin{bmatrix} s\,\mathbf{R} & \mathbf{t} \\ \mathbf{0}^\top & 1 \end{bmatrix} \mathbf{T}^{ee}_{rob}(t). \tag{7}$$

The gripper opening state $c_t$, extracted from the joint configuration, together with $\mathbf{T}^{ee}_{sim}(t)$, defines the robot action used to drive object interactions in simulation.

**Physical reference frames.** The physical reference direction confirmed by fitting the supporting table plane $\Pi$ from the reconstructed point cloud. Let $\mathbf{n}_\Pi$ denote the plane normal and $\mathbf{v}_c$ the camera viewing direction. The gravity is defined as:

$$\mathbf{g} = -\,\mathrm{sign}(\mathbf{n}_\Pi \cdot \mathbf{v}_c)\,\mathbf{n}_\Pi, \tag{8}$$

which resolves the sign ambiguity of gravity and aligns the simulation with the real-world scene.

### 4.2.2. FORCE-DRIVEN GS DYNAMICS MODELING

Existing neural GS dynamics models are primarily state-based or model interactions only weakly, which is sufficient for isolated or simply interacting deformable objects. In robot manipulation, however, object motion is largely governed by contact-induced forces from the robot and the environment. Such neural models often fail to generalize across diverse interaction patterns, as they entangle object motion with contact-specific effects.

Therefore, we model environment interactions as explicit global forces and robot interactions as implicit forces computed through the interaction graph, both applied directly at the Gaussian splat (GS) level. These forces are hierarchically propagated through a neural network, where each level predicts the next state conditioned on both the current state and interaction forces, enabling force-driven object dynamics rather than state-only regression.

**Force-driven dynamics.** For each GS node or cluster $i$, the linear velocity and rotation are predicted from its historical states and the aggregated interaction force:

$$(\mathbf{v}_i, \boldsymbol{\omega}_i) = \psi_\theta\left(g_i^{t-1}, g_i^{t-2}, \mathbf{f}_i, dt\right), \tag{9}$$

where $\psi_\theta$ denotes a hierarchical neural dynamics model and $\mathbf{f}_i$ is the total interaction force acting on node $i$.

**Environment force.** Environmental effects are modeled as external forces. Each GS node is subject to gravity, and nodes close to the supporting surface receive an additional support force:

$$\mathbf{f}_i^{\text{env}} = \begin{cases} \mathbf{g} + \mathbf{s}_i, & d_i < \tau, \\ \mathbf{g}, & d_i \geq \tau, \end{cases} \tag{10}$$

where $d_i$ denotes the signed distance from the Gaussian splat to the supporting plane. Environment forces are aggregated bottom-up across the hierarchical structure.

**Robot force.** Robot interaction is modeled via an interaction graph constructed between robot control points and GS nodes. Given the robot action at time $t$, the robot-induced force on GS node $i$ is predicted as:

$$\mathbf{f}_i^{\text{rob}} = \Phi_\theta\left(g_i^{t-1}, \{\mathbf{r}_j^t\}_{j\in\mathcal{N}(i)}, c_t\right), \tag{11}$$

where $\{\mathbf{r}_j^t\}$ are neighboring robot control points, $c_t$ denotes the gripper state, and $\Phi_\theta$ is a graph-based neural interaction module. The total interaction force is given by $\mathbf{f}_i = \mathbf{f}_i^{\text{env}} + \mathbf{f}_i^{\text{rob}}$.

### 4.2.3. MULTI-RESO. TRAINING FOR LONG HORIZON

Learning GS-based dynamics for embodied manipulation is particularly challenging over long horizons. Small prediction errors at the splat level accumulate rapidly under repeated robot interactions, leading to drift and instability in both geometry and appearance. Moreover, GS-based simulation is conditioned on a specific initial splat configuration $G_0$, which prevents random temporal cropping or noise-based reinitialization commonly used in particle-based simulators. To address these challenges, we adopt a multi-resolution training strategy along both temporal and image dimensions to enable efficient and stable long-horizon learning.

**Multi-resolution temporal.** We employ a coarse-to-fine temporal training scheme. In the first stage, the model is trained with a larger temporal stride $k \cdot dt$ to capture long-range dynamics. In the second stage, training is performed at the original resolution $dt$ using randomly sampled subsequences of length $n \cdot k$, enabling fine-grained dynamics learning while mitigating error accumulation.

**Multi-resolution image.** GS geometry is reconstructed from super-resolution images to preserve structural details, while dynamics training is conducted at original image resolution to reduce computational cost.

### 4.2.4. IMMIGRATED SUPERVISION

Embodied manipulation inherently suffers from partial observability, as deformable objects are frequently occluded by the robot end-effector or by self-contact during interaction. This prevents the direct use of reliable 3D tracking as supervision and makes pure image-based losses insufficient for recovering occluded object states. To address this challenge, we adopt a blended training strategy that combines occlusion-aware image supervision for visible regions with physics-inspired consistency constraints to regularize unobserved dynamics.

**Occlusion-aware image supervision.** Naively supervising rendered images over the full image domain introduces spurious gradients from occluded regions and leads to unstable dynamics learning. We therefore apply image-based supervision selectively to visible object regions, allowing reliable visual evidence to guide GS-level state updates where observations are available.

Given a binary object mask $\mathbf{M}_t$, the image reconstruction loss is defined as:

$$\begin{aligned} \mathcal{L}_{\text{img}}^{(t)} = {} & \lambda \left\| \mathbf{M}_t \odot (\hat{\mathbf{I}}_t - \mathbf{I}_t) \right\|_2^2 \\ & + (1-\lambda)\, \mathcal{L}_{\text{D-SSIM}}\left(\mathbf{M}_t \odot \hat{\mathbf{I}}_t, \mathbf{M}_t \odot \mathbf{I}_t\right), \end{aligned} \tag{12}$$

where $\odot$ denotes element-wise masking and the loss is evaluated only within the masked region.

**Physics consistency regularization.** While image supervision constrains dynamics in visible regions, occluded splats lack direct visual feedback during training. To prevent physically inconsistent motion in these unobserved regions, we introduce a physics consistency regularization over the hierarchical GS representation. This constraint propagates physically plausible dynamics across hierarchy levels and mitigates long-horizon drift caused by accumulated prediction errors.

Specifically, we enforce mass conservation between adjacent hierarchy levels via:

$$\mathcal{L}_{\text{mom}} = \sum_{l=1}^{L-1} \left\| m_{c_l}\hat{\mathbf{x}}_{c_l} - \sum_{i\in\mathcal{C}_{l-1}} m_i\hat{\mathbf{x}}_i \right\|_2^2, \tag{13}$$

where $\mathcal{C}_{l-1}$ denotes the set of child nodes of cluster $c_l$. This term provides a self-supervised physical regularization without requiring ground-truth forces or contacts.

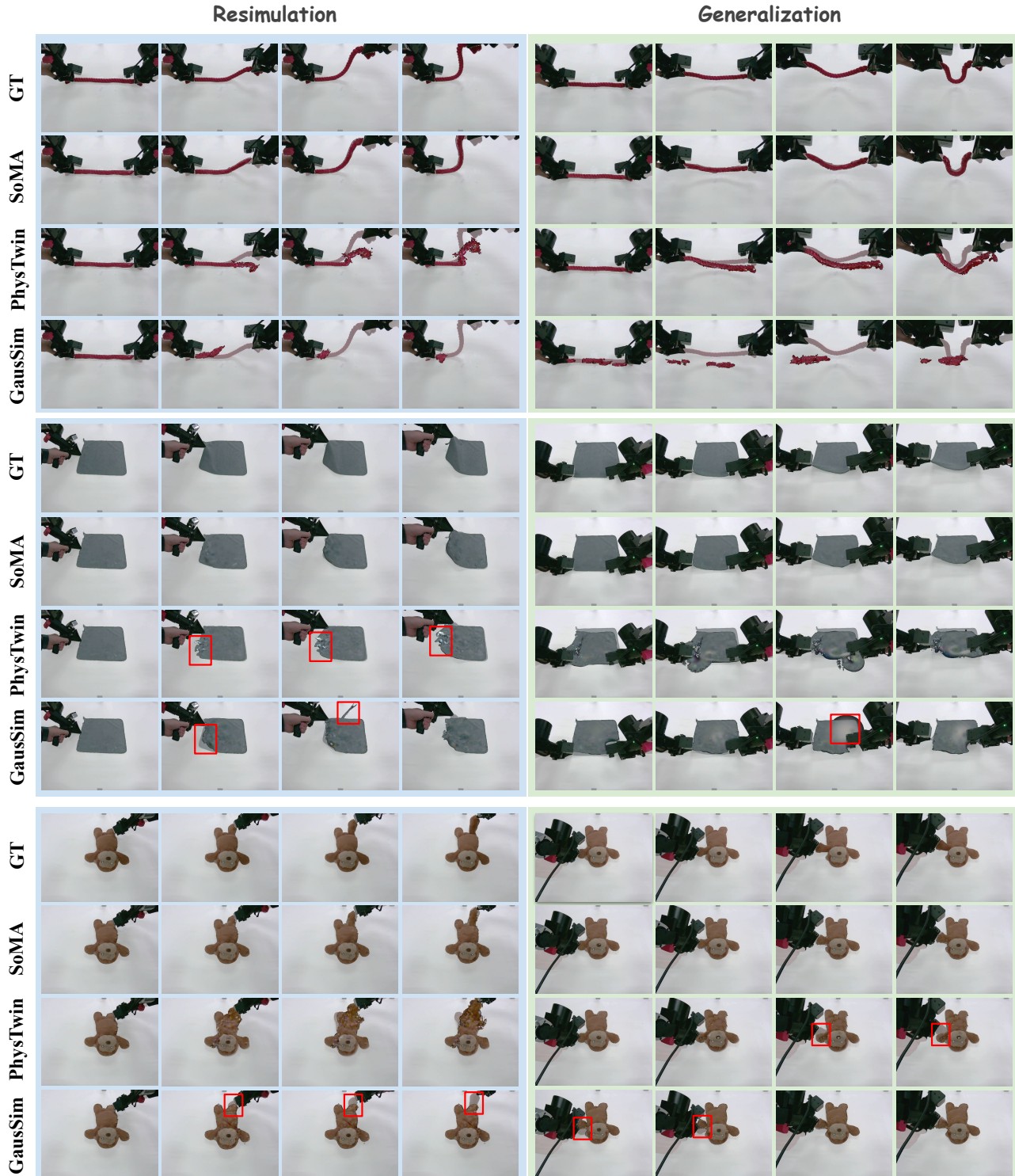

*Figure 3.* **Qualitative resimulation and generalization under robot manipulation.** Left: resimulation on training trajectories. Right: generalization to unseen robot actions and contact configurations. Across diverse soft-body objects, including near-linear (rope), near-planar (cloth), and volumetric (doll) objects, SoMA produces stable, long-horizon simulations that closely match observed dynamics. PhysTwin shows deviations under complex or unseen interactions due to real-to-sim mismatch, while GausSim often remains static or unstable in challenging scenarios.

# 5. Experiments

## 5.1. Experimental Settings

**Datasets.** We collect real-world robot manipulation datasets on an ARX-Lift platform, covering four deformable objects: *rope*, *doll*, *cloth*, and *T-shirt*. RGB images are captured at a resolution of $640 \times 480$ and 30 FPS, with robot joint states recorded synchronously. Most sequences contain 100–150 frames. For each object, we collect 30–40 sequences with diverse initial configurations and manipulation actions. The datasets are split into training and test sets with a ratio of 7:3.

**Tasks.** We evaluate models on two tasks: (i) **Resimulation**, which measures reconstruction and simulation accuracy on training trajectories; (ii) **Generalization**, which evaluates performance on unseen manipulation sequences in the test set. In both settings, models are initialized with reconstructed Gaussian splats and perform open-loop simulation conditioned on per-frame robot actions.

**Baselines.** As no existing method directly supports robot-manipulated scenes with complex interactions and occlusions, we compare against representative physics-based and neural approaches, including **PhysTwin** (Jiang et al., 2025a) and **GausSim** (Shao et al., 2025). All baselines are adapted to our setting for fair comparison (details in Appendix B.3).

**Evaluation metrics.** We evaluate simulation quality using observation-based metrics. For RGB observations, we report PSNR, SSIM, and LPIPS (Wang et al., 2004; Wang, 2004; Zhang et al., 2018) to measure pixel-level fidelity, structural similarity, and perceptual consistency, respectively. Since full 3D ground truth is unavailable in real-world manipulation scenes, we use depth as a geometric proxy and evaluate depth accuracy using Absolute Relative Error (Abs Rel) and RMSE on valid regions. Abs Rel captures scale-invariant relative depth errors, while RMSE reflects absolute geometric deviations. All metrics are averaged over all frames within each video sequence and across evaluation scenarios.

## 5.2. Results

We present experimental results to evaluate the real-to-sim simulation capability of our method under progressively more challenging settings. First, we assess both resimulation and generalization performance on cloth, rope, and doll with diverse manipulation actions. Next, to examine the potential of our approach in embodied manipulation scenarios, we evaluate it on the more challenging T-shirt folding dataset, which involves complex interactions and large deformations. Finally, we provide ablation studies to analyze the contribution of key components in our framework.

**Resimulation performance across deformable objects.** Fig. 3 shows that our method produces resimulated trajectories that closely match ground-truth observations over long temporal horizons. The simulated object dynamics remain stable throughout the rollout, accurately preserving both global motion and local deformations.

Quantitative results in Tab. 1 further confirm the robustness of our approach, where our method achieves the best performance across all metrics on the resimulation task. Compared to the differentiable simulator PhysTwin, which relies on explicit point tracking and degrades under occlusion-heavy robot manipulation, our method yields more accurate dynamics and more consistent visual reconstructions. In contrast, state-based neural simulators such as GausSim exhibit increasing deviation in later timesteps, failing to preserve long-term object dynamics, particularly under interaction. Overall, these results demonstrate that our method enables accurate and robust resimulation of deformable objects in complex robot manipulation scenarios.

**Generalization to unseen manipulations.** We evaluate generalization under manipulation settings that differ from those observed during training, including novel action trajectories and contact configurations. As shown in Fig. 3, our method maintains accurate and stable dynamic evolution in such settings, closely matching the ground-truth observations. Quantitative results in Tab. 1 further confirm strong generalization performance.

A key advantage of our approach is its controllability: by conditioning dynamics on robot actions and interaction cues, the simulator can produce consistent object behavior beyond memorized trajectories. In contrast, state-based simulators exhibit large deviations when exposed to unseen actions, while differentiable physics-based methods often fail to preserve coherent object structure under complex interactions, resulting in inaccurate deformations. Overall, these results demonstrate the robustness of SoMA in generalizing real-to-sim simulation across unseen manipulation scenarios.

**Complex interaction: T-shirt folding.** *T-shirt* folding involves long-horizon dynamics, large deformations, and frequent self-contacts, making it substantially more challenging than prior settings. Such complexity amplifies error accumulation and often leads to structural collapse or severe artifacts in existing simulators.

Fig. 1 shows that our method can stably simulate the full folding process with coherent geometry and realistic dynamics. (more results shows in Appendix C.1). In contrast, PhysTwin exhibits pronounced artifacts and inconsistent deformations under the same setting, failing to preserve a valid object structure throughout the rollout. Quantitatively, our method consistently outperforms baselines across all

*Table 1.* **Quantitative evaluation on resimulation and generalization under robot manipulation.** We report performance comparisons with PhysTwin and GausSim across image-based and depth-based metrics. Our method achieves the best results across all metrics, demonstrating robust and accurate real-to-sim simulation.

| Task | Resimulation | | | | | Generalization | | | | |
|---|---|---|---|---|---|---|---|---|---|---|
| Method | Abs Rel↓ | RMSE↓ | PSNR↑ | SSIM↑ | LPIPS↓ | Abs Rel↓ | RMSE↓ | PSNR↑ | SSIM↑ | LPIPS↓ |
| PhysTwin (Jiang et al., 2025a) | 0.102 | 0.150 | 28.77 | 0.947 | 0.086 | 0.128 | 0.168 | 26.54 | 0.941 | 0.092 |
| GausSim (Shao et al., 2025) | 0.115 | 0.155 | 31.69 | 0.945 | 0.092 | 0.143 | 0.186 | 31.29 | 0.942 | 0.127 |
| **SoMA (ours)** | **0.089** | **0.124** | **33.51** | **0.971** | **0.055** | **0.112** | **0.137** | **32.89** | **0.968** | **0.062** |

*Table 2.* **Quantitative results on the *T-shirt* folding task under robot manipulation.**

| Method | Abs Rel↓ | RMSE↓ | PSNR↑ | SSIM↑ | LPIPS↓ |
|---|---|---|---|---|---|
| PhysTwin | 0.129 | 0.208 | 22.85 | 0.842 | 0.198 |
| **SoMA** | **0.112** | **0.172** | **27.57** | **0.896** | **0.128** |

*Table 3.* **PSNR Ablation on *cloth* dataset (150-frame sequences).** We evaluate SoMA with different components: blended supervision (Sec. 4.4), image-only supervision (`Img-only`), and without multi-resolution training (`w/o MRT`, Sec. 4.3). `Jointly`: training on all domains to test generalization.

| | Full Model | Jointly | Img-Only | w/o MRF |
|---|---|---|---|---|
| Resim | **32.73** | 32.59 | 30.29 | 31.97 |
| Genral | 31.49 | **31.77** | 29.17 | 30.29 |

metrics, reflecting both improved stability and accuracy.

These results demonstrate that SoMA can handle task-level soft-body manipulation with complex interactions, highlighting its potential as a practical simulation tool for embodied manipulation scenarios.

**Ablation studies.** We conduct ablation studies on the *cloth* domain using PSNR as the evaluation metric, as summarized in Tab. 3. We compare the full model with three variants: **Jointly**, which is trained on all object domains; **Img-only**, which removes the blended supervision loss (Sec. 4.4) and uses raw image supervision only; and **w/o MRT**, which disables the multi-resolution training strategy (Sec. 4.3).

The full model achieves the best overall performance. Joint training slightly reduces resimulation accuracy but improves generalization performance, indicating its potential for learning more robust dynamics from diverse data and scaling to larger manipulation datasets. Removing multi-resolution training leads to a consistent drop in performance, confirming its role in stabilizing long-horizon simulation. Notably, the **Img-only** variant suffers the largest degradation, especially in generalization, highlighting the importance of the blended supervision with masked image loss and physical constraints for learning transferable interaction dynamics.

## 6. Applications

SoMA enables forward simulation of deformable objects under rich robot–object interactions, providing a practical virtual environment for analyzing complex soft-body behaviors. Its stable long-horizon simulation and direct robot-action conditioning support prediction of object dynamics under different manipulation strategies. As a real-to-sim backend with a reduced reality gap, SoMA facilitates simulation-driven robot learning and improves policy transfer to real-world manipulation. For example, it can stably simulate long-horizon, self-contact–heavy tasks such as T-shirt folding, enabling task-level analysis and policy development.

## 7. Conclusion

We presented **SoMA**, a neural real-to-sim simulator for soft-body manipulation in robot interaction scenes. Operating directly on Gaussian splat representations, our approach learns deformable object dynamics end-to-end from RGB observations without relying on predefined physical models. By explicitly modeling robot–object interactions, the proposed simulator enables accurate and stable long-horizon simulation conditioned on robot actions. Experiments demonstrate strong generalization to unseen interactions and consistent improvements over prior methods across diverse objects and complex manipulation tasks such as cloth folding.

## Acknowledgments

This work is funded in part by the National Key R&D Program of China (2022ZD0160201), and Shanghai Artificial Intelligence Laboratory. This work was supported by the National Natural Science Foundation of China (Grant No. 62502247). This research was partially supported by the HKU Startup Fund and Institute of Data Science.

## Impact Statement

Existing rule-based simulators rely on manually specified physical parameters, while state-based simulators provide limited action-conditioned interaction modeling. SoMA may reduce real-robot data collection costs by enabling more reliable R2S simulation of deformable objects.

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

# A. Discussion

## A.1. Distinguishing Different Technical Paradigms

Recent advances in dynamic scene modeling and simulation have led to increasingly similar visual results across different methods, which may obscure their fundamental differences. Despite comparable visual quality, these approaches are motivated by distinct problem settings and adopt different design trade-offs.

**Passive Reconstruction vs. Interaction-Aware Simulation.** 4D reconstruction methods (Singer et al., 2023a; Bahmani et al., 2025; Borycki et al., 2026) focus on recovering temporally consistent geometry and appearance from visual observations, without explicitly modeling physical states or interactions. Consequently, they are not suitable for controllable manipulation or physical reasoning.

**Video World Models vs. Physics-Oriented Simulation.** Video World models (Huang et al., 2026; Li et al., 2025; Yu et al., 2025) are typically trained on large-scale video or interaction data and excel at predicting perceptually plausible future observations. However, they are optimized for appearance realism rather than physical or geometric consistency, which can lead to view inconsistency and physically implausible behaviors under external interventions. These characteristics limit their applicability to tasks requiring precise physical reasoning or controllable interaction.

**Rule-Based Simulation vs. Data-Driven Dynamics.** Rule-based simulators (Jiang et al., 2025a; Zhang et al., 2024b) rely on predefined physical formulations and parameters, including simplified object properties for differentiability and simulator-specific parameters that require manual tuning. Such design choices constrain their expressiveness and adaptability in real-world settings. In contrast, data-driven neural simulators encode physical properties using latent embeddings and learn interaction dynamics directly from data, enabling better adaptability across diverse objects and scenes without explicit parameter reconfiguration.

**Position of SoMA.** SoMA is designed for robot manipulation under real-to-sim settings. By explicitly representing physical states while learning interaction-aware dynamics from data, it occupies a distinct point in the design space that balances physical consistency, controllability, and fidelity to real-world observations.

*Table 4.* Comparison of different technical paradigms for dynamic scene modeling and simulation.

|  | 4D Recon. | Video WM | Rule-based Sim. | Neural Sim. | SoMA |
|---|---|---|---|---|---|
| Explicit Physical State | ✗ | ✗ | ✓ | ✓ | ✓ |
| Action-conditioned | ✗ | ✗ | ✓ | ✗ | ✓ |
| View/3D Consistency | ✓ | ✗ | ✓ | ✓ | ✓ |
| Real-to-Sim Applicability | ✗ | ✓ | ✗ | ✓ | ✓ |

## A.2. Distinguishing SoMA from Related R2S works

Complementary to the paradigm-level discussion above, Tab. 5 provides a method-level comparison between SoMA and closely related baselines. Among them, PhysTwin(Jiang et al., 2025a) and GS-Dynamics(Zhang et al., 2024a) are the most relevant to robot-conditioned interaction, as they also study dynamics from real observations. However, they differ from SoMA in problem setting and supervision: PhysTwin focuses on hand manipulation, while GS-Dynamics relies on additional 3D tracking/control-point supervision. GausSim(Shao et al., 2025) and 3DGSim(Zhobro et al., 2025) are also GS-based visual simulation methods, but they mainly target deformation or collision/deformation from multi-view observations rather than robot-action-conditioned manipulation.

VPD/HD-VPD(Whitney et al.; 2025), PointWorld(Huang et al., 2026), and DeformContact(Saleh et al., 2024) instead emphasize geometry-driven dynamics with different 3D representations, including latent particles, point clouds, and meshes. These representations provide useful physical or geometric structure, but their rendering capabilities differ from GS-based methods: PointWorld and DeformContact do not report rendering results, while VPD/HD-VPD produces rendered outputs with relatively coarse visual fidelity. Overall, SoMA is distinguished by its focus on real-world robot manipulation, action-conditioned dynamics, multi-view training supervision, and GS-based rendering quality.

*Table 5.* Comparison with related dynamic scene modeling and simulation methods.

| Method | Data | Dynamic Pattern | Action-cond. | 3D Representation | Supervision | Rendering |
|---|---|---|---|---|---|---|
| SoMA (ours) | **Real** | **Robot Manip** | **yes** | Gaussian Splats | Multi-view Videos | **yes** |
| PhysTwin | **Real** | Hand Manip | **yes** | Gaussian Splats & Point Cloud | Multi-view Videos & 3D tracking | **yes** |
| GS-Dynamics | **Real** | **Robot Manip** | **yes** | Gaussian Splats & Control Points | Multi-view Videos & 3D tracking | **yes** |
| GausSim | **Real** | Deformation | no | Gaussian Splats | Multi-view Videos | **yes** |
| 3DGSim | Virtual | Collision & Deform | no | Gaussian Splats | Multi-view Videos | **yes** |
| VPD/HD-VPD | **Real** | **Robot Manip** | **yes** | Latent Particles & Point Cloud | Multi-view Videos | Coarse |
| PointWorld | **Both** | **Robot Manip** | **yes** | Point Cloud | 3D tracking | no |
| DeformContact | Virtual | Deformation | no | Mesh | / | no |

## A.3. Limitations and Future Directions

While our experiments demonstrate that SoMA can generalize across multiple object categories and diverse manipulation actions, the current evaluation is still limited in scale. Specifically, our datasets include four object categories with a range of interaction types, and generalization is evaluated on unseen actions within this scope. Although these results are encouraging, broader validation is required to assess scalability and robustness under more diverse object geometries, materials, and interaction patterns.

A natural future direction is to extend the training and evaluation to significantly larger and more diverse datasets, potentially covering hundreds of object instances and a wider spectrum of manipulation behaviors. Such large-scale studies would help better characterize the limits of data-driven interaction-aware simulation and examine whether generalization continues to improve with increased data diversity.

## B. Implementation Details

### B.1. Dataset Collection & Preprocessing

We collect real-world manipulation data using the `arx_lift` platform with three Intel RealSense D405 RGB cameras. The data collection setup is illustrated in Fig. 1. One camera is rigidly mounted on the robot end-effector and serves as a key component for robot-conditioned real-to-sim mapping, while the other two cameras are placed around the tabletop to provide multi-view observations. All RGB images and robot joint states are synchronously recorded at 30 FPS.

**Image Processing.** All captured images are segmented using GroundingDINO (Liu et al., 2024) and Grounded-SAM2 (Ravi et al., 2025) with text prompts. We generate two types of masks: object masks, which provide the primary visual observations for end-to-end supervision, and robot masks, which are used to identify occluded regions caused by the manipulator during interaction.

**Robot State Processing.** Given the robot joint states, we compute the end-effector pose and gripper opening parameters in the robot base frame using the provided URDF model. These quantities are later mapped into a unified simulation coordinate system via the robot-conditioned real-to-sim transformation.

**Camera Pose Estimation and Reconstruction.** To establish the reconstruction coordinate system, we select frames with rich visual texture as references and apply segmentation and super-resolution preprocessing (Rombach et al., 2022). Camera poses and an initial point cloud are reconstructed using Pi3 (Wang et al., 2025b). The resulting point cloud is further optimized and converted into a 3D Gaussian Splatting (3DGS) (Kerbl et al., 2023) representation for simulation and rendering. For the T-shirt folding scenes, we initialize all Gaussian splat colors to blue to ensure consistent appearance for splats corresponding to the back side of the cloth, which may be weakly observed or fully occluded.

**Transformation of Robot Conditioning.** As defined in Eq. 6, the robot-to-simulation mapping is determined by a scale factor $s$ and a rigid transformation $(\mathbf{R}, \mathbf{t})$. The scale $s$ is computed by matching the physical size of reference objects across coordinate systems. The rigid transformation is resolved via the mounted camera, whose camera-to-world pose is known in both the robot frame and the reconstruction frame. Specifically, let $\mathbf{T}_{\text{cam}}^{\text{rob}}$ denote the camera-to-world transformation in the robot frame obtained from the URDF, and $\mathbf{T}_{\text{cam}}^{\text{rec}}$ the corresponding pose in the reconstruction frame estimated by Pi3. The robot-to-simulation transformation is then given by

$$\mathbf{T}_{\text{rob}\rightarrow\text{sim}} = \begin{bmatrix} \mathbf{R} & \mathbf{t} \\ \mathbf{0}^\top & 1 \end{bmatrix} = \mathbf{T}_{\text{cam}}^{\text{rec}} \left(\mathbf{T}_{\text{cam}}^{\text{rob}}\right)^{-1}. \tag{14}$$

### B.2. Model Implementation

**Implementation Efficiency.** All experiments are conducted on NVIDIA H200 GPUs. We use four GPUs for training and a single GPU for inference. The first-stage training takes approximately 24 hours.

**Hierarchical Clustering.** We organize Gaussian splats into a fixed three-level hierarchy (including the finest splat level) across all datasets. The hierarchy is constructed once using distance-based clustering and remains unchanged during training and inference. To satisfy the input requirements of the hierarchical network, we adopt a fixed clustering scheme with approximately $[n, n/2, 2]$ nodes from fine to coarse levels, where $n$ denotes the number of Gaussian splats. The number of control points is fixed to 30. Detailed clustering parameters for each dataset are provided in Tab. 6.

*Table 6.* **Quantitative results on the cloth folding task under robot manipulation.**

|  | rope | cloth | doll | T-shirt |
|---|---|---|---|---|
| downsample_rate | [0.02, 0.2] | [0.02, 0.2] | [0.02, 0.2] | [0.02, 0.2] |
| num of splats | 13000 | 13000 | 13000 | 13000 |
| num of clusters | [8, 800] | [30, 2400] | [22, 2200] | [60, 3000] |

**Network Architecture.** Our simulator backbone is a hierarchical mesh graph network. Each hierarchy level employs a mesh graph network with shared parameters, enabling consistent multi-scale message passing. All graph networks use an embedding dimension of 128 and consist of 16 encoder layers with SiLU activations and normalization applied at each layer. The propagated splat features are decoded by a lightweight MLP head to predict per-splat state updates, including velocity, deformation, and rotation.

**Multi-Temporal Training.** To improve training stability and long-horizon rollout performance, we adopt a multi-resolution temporal training strategy. Instead of supervising dynamics at the original frame rate throughout training, we first train the simulator using temporally subsampled sequences with an interval of $k$ frames. Specifically, for the T-shirt folding dataset we use $k = 5$, as this task involves larger deformations and more complex motions, requiring finer temporal resolution during coarse training. For all other datasets, we use $k = 10$. This coarse temporal supervision encourages the model to capture global motion patterns and interaction effects over longer time spans.

After the initial stage, the model is fine-tuned using full-resolution sequences with the original frame interval. An ablation study on the choice of $k$ is provided in Tab. 7, showing that the performance is relatively insensitive to the exact value of $k$ within a reasonable range.

**Image Super-Resolution.** To obtain higher-quality Gaussian splat reconstructions, we apply a $4\times$ image super-resolution preprocessing to selected frames before segmentation and reconstruction. The enhanced image resolution provides sharper object boundaries and more accurate masks, which in turn improves the quality of the reconstructed point clouds and the resulting 3D Gaussian splats.

### B.3. Baselines Implementation

**PhysTwin.** For PhysTwin (Jiang et al., 2025a), the objects in our datasets are similar to those used in the original PhysTwin benchmark. We therefore follow the vanilla PhysTwin setting whenever applicable. To enable robot manipulation, we represent the robot end-effector as a circular control point with a fixed radius, which serves as the interaction primitive for

*Table 7.* Ablation Study on cloth_lift.

| $k$ | 5 | 10 | 15 | 20 | 30 |
|---|---|---|---|---|---|
| PSNR↑ | 31.87 | 32.73 | 32.57 | 32.53 | 32.10 |
| RMSE↓ | 0.120 | 0.126 | 0.126 | 0.123 | 0.122 |

grasping and contact. This modification only adapts the control abstraction and does not alter the underlying physical model of PhysTwin.

**GausSim.** GausSim (Shao et al., 2025) does not support explicit control or action inputs and instead predicts future Gaussian splat states in a purely state-based autoregressive manner. The model estimates the previous state $\mathbf{GS}_{t-1}$ from the initial splats and regresses the full sequence accordingly. For the resimulation setting, we train GausSim on full sequences following its original protocol. For the generalization setting, we provide $\mathbf{GS}_0$ and $\mathbf{GS}_1$ as initial conditions and directly evaluate its rollout performance without additional control inputs.

### B.4. Evaluation Metrics

**Image Metrics.** We evaluate rendered image quality using PSNR, SSIM, and LPIPS (Wang et al., 2004; Wang, 2004; Zhang et al., 2018). All image metrics are computed between the predicted images and the ground-truth RGB images within the object regions only. Specifically, object masks are applied to exclude background pixels and occluded regions, ensuring that the evaluation focuses on the visible object appearance.

**Depth Metrics.** To assess geometric accuracy without full 3D ground truth, we report depth-based metrics including Absolute Relative Error (Abs Rel) and RMSE. Depth metrics are computed on valid object regions. For masked-out pixels corresponding to the background or occluded areas, we assign the depth value of the supporting tabletop to maintain consistent depth maps and avoid undefined values during metric computation.

## C. More Results

### C.1. Multi-view Results

Multi-view qualitative results are shown in Fig. 4(a). Our method maintains consistent visual accuracy and physically plausible dynamics across both the main view and the side view, demonstrating robustness to viewpoint changes.

### C.2. T-shirt Folding Comparison Results

Qualitative comparison results for the T-shirt folding task are shown in Fig. 4(b). Our method accurately simulates the folding process, while PhysTwin exhibits noticeable deviations from the ground-truth motion.

### C.3. Results on the Datasets of PhysTwin

*Table 8.* **Quantitative results on the Datasets of Phystwin.**

| | PhysTwin (avg) | **SoMA (avg)** | SoMA (rope) | SoMA (cloth) | SoMA (doll) |
|---|---|---|---|---|---|
| PSNR↑ | 28.214 | 32.640 | 34.261 | 32.739 | 30.920 |
| SSIM↑ | 0.945 | 0.981 | 0.988 | 0.981 | 0.973 |
| LPIPS↓ | 0.034 | 0.026 | 0.018 | 0.032 | 0.029 |

To evaluate SoMA under a different experimental setting, we apply it to the PhysTwin datasets(Jiang et al., 2025a). PhysTwin

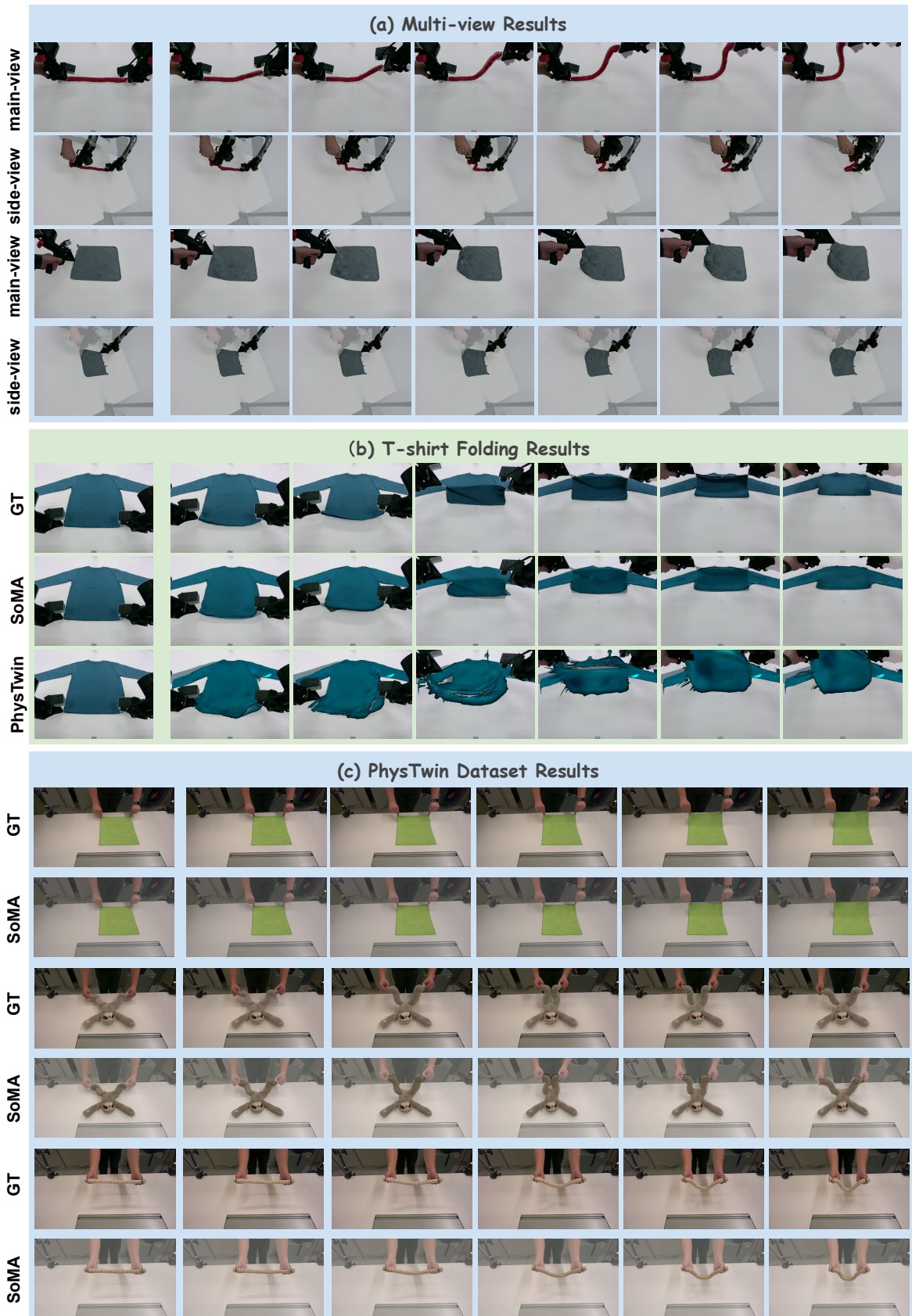

*Figure 4.* **More Results** (a) multi-view results; (b) T-shirt folding comparison results (c) results on PhysTwin datasets.

provides 3D-tracked hand trajectories as control signals instead of explicit robot actions; we therefore treat these trajectories as robot end-effector motions and assume a closed gripper during interaction.

Qualitative results are shown in Fig. 4(c). Our method produces dynamics that are visually and physically consistent with the ground-truth sequences under this setting. Quantitative results are reported in Tab. 8, where SoMA consistently outperforms baseline methods across all evaluated metrics. These results demonstrate that our simulator generalizes beyond robot-action-driven settings and can effectively model interaction dynamics even when driven by externally provided motion trajectories.

