# OpenReview forum: "SoMA: A Real-to-Sim Neural Simulator for Robotic Soft-Body Manipulation"
_ICML.cc/2026/Conference — ICML 2026 regular_

### Official Review · Reviewer_GWXV · 2026-03-06

**Soundness:** 3
**Presentation:** 3
**Significance:** 3
**Originality:** 3
**Overall Recommendation:** 5
**Confidence:** 3

**Summary:**

The paper proposes a real-to-sim neural simulator for robot manipulation of deformable objects. The approach represents deformable objects using 3D Gaussian splats and learns action-conditioned dynamics directly from multi-view RGB observations and robot joint states. Unlike traditional physics-based simulators that rely on predefined physical models, SoMA learns object dynamics in a unified latent representation that incorporates robot actions, environmental effects, and deformable object states.

The system is trained end-to-end from real-world robot interaction data and performs open-loop simulation conditioned on robot actions, enabling resimulation of observed manipulation trajectories as well as generalization to unseen actions. The authors evaluate the approach on real-world datasets involving deformable objects such as rope, cloth, dolls, and T-shirts. Experimental results show improvements over adapted baselines (PhysTwin and GausSim) and demonstrate stable long-horizon simulation.

**Compliance With Llm Reviewing Policy:**

Affirmed.

**Final Justification:**

The rebuttal addressed the main concerns I raised, and I have therefore adjusted my recommendation to accept.

**Key Questions For Authors:**

1. Simulation fidelity beyond visual metrics
The evaluation focuses mainly on image and depth reconstruction metrics. Can the authors provide additional evaluation of physical accuracy, such as trajectory error, deformation consistency, or contact dynamics?

2. Scalability to larger datasets and tasks
The current datasets are relatively small. How does the method scale with larger datasets, longer trajectories, or more complex manipulation tasks?

3. Simulation efficiency
What are the training time, inference speed, and memory requirements of the proposed simulator? Could the system realistically be used for large-scale policy learning or simulation-based planning?

4. Generalization limits
How does the model perform under out-of-distribution manipulations, such as stronger forces, unseen object configurations, or longer rollouts?

5. Ablation on Gaussian representation choice
How critical is the use of Gaussian splats compared to other representations (e.g., point clouds, particles, neural fields)? An ablation comparing representations would clarify the contribution.

**Limitations:**

Consider adding a short section discussing limitations related to dataset size and generalization, challenges in modeling complex deformable physics and potential misuse or risks associated with realistic simulation tools.

**Strengths And Weaknesses:**

Strengths:

1. The paper addresses an important challenge in robotics and machine learning: real-to-sim modeling for deformable object manipulation. Accurate simulation of deformable objects remains difficult due to complex dynamics and interactions, and improving real-to-sim transfer is highly relevant for robot learning and data-efficient policy training.

2. The explicit conditioning of the simulator on robot joint actions allows the system to model causal robot–object interactions rather than simply reconstructing motion trajectories. This is an important step beyond many existing neural reconstruction approaches that primarily replay observed motion.

3. Using Gaussian splats as the simulation representation is interesting and potentially powerful. Operating directly on Gaussian splats for dynamics simulation is a novel and promising design choice.

4. The work includes real-world robot manipulation datasets with multiple deformable objects. Demonstrating real-to-sim modeling on real-world data (rather than synthetic benchmarks) strengthens the practical relevance of the work.

5. The evaluation distinguishes between resimulation (reconstructing training trajectories) and generalization to unseen manipulations, which is a useful distinction when assessing learned simulators.

Weaknesses

1. Although gathering real-world data is difficult at scale, the experimental evaluation appears relatively limited:

Each object dataset contains only 30–40 sequences.
Sequence length is relatively short (100–150 frames).
Only four objects are evaluated.

2. The primary evaluation metrics are PSNR, SSIM, LPIPS, and depth errors, which mainly measure visual reconstruction fidelity. However, a manipulation simulator should ideally also be evaluated on physical plausibility or downstream utility, such as accuracy of predicted object motion trajectories, contact dynamics consistency, usefulness for training manipulation policies.
Without such evaluation, it is difficult to determine whether the simulator captures true physical dynamics or mainly reproduces visually plausible motion.

3. The paper compares against only two baselines (PhysTwin and GausSim). While the authors note that few existing methods directly address this setting, the comparison still feels somewhat narrow.
A broader comparison would strengthen the empirical claims.

4. Although the paper claims improved generalization to unseen manipulations, the experiments appear limited to test sequences from the same distribution of objects and actions. Stronger generalization tests could include unseen object instances and more complex manipulations.

5. The paper claims stable long-horizon simulation, but the evaluation does not systematically analyze rollout stability over very long horizons or error accumulation.More detailed analysis would help validate this claim.

6. The paper does not clearly report training time, inference cost, simulation speed relative to physics-based simulators. Since real-to-sim simulators are often used in large-scale data generation or RL training, computational efficiency is an important factor.

---

> ### Author Rebuttal · Authors · 2026-03-31
>
> Anonymous Link (Fig.5): https://github.com/anonymousrebuttal2000/anonymous_figure
>
> ### W1 & W4 & W5 & Q4 Evaluation Span / Generalizability
>
> Although each real-world object dataset contains only 30–40 sequences, SoMA shows strong generalization from limited data. Beyond the results in Fig. 3 and Fig. 4, the additional experiment in Fig.5 (b) shows transfer across different actions, textures, patterns, sizes, materials, and manipulators. For example, a model trained on a small blue cloth generalizes to larger cloths with different visual patterns and material properties, indicating that the method does not simply overfit the small training set. These additional results also cover more challenging and out-of-distribution settings, including unseen object variants and stronger manipulations.
>
> For long-horizon stability, our setting is already more challenging than many prior works in both rollout length and deformation magnitude. In addition to the qualitative results in Fig. 3 and Fig. 4, we further provide step-wise analysis of tracking error and PSNR at rollout steps 20/50/80/100 (shown below Tab. 10), which directly measures error accumulation over time. The results show that SoMA remains stable even at later rollout stages, supporting our long-horizon simulation claim.
>
> **Table 10. Step-wise trajectory error comparison across rollout horizons.**
> | rollout     | 20 steps   | 50 steps   | 80 steps   | 100 steps  |
> |----------|-------|-------|-------|-------|
> | PhysTwin | 0.022 | 0.039 | 0.061 | 0.072 |
> | GausSim  | 0.027 | 0.042 | 0.052 | 0.059 |
> | SoMA (ours)    | **0.008** | **0.012** | **0.015** | **0.016** |
>
> ### W2 & Q1 Evaluation Metrics
>
> Depth errors already provide a physics-related signal in our setting, since ground-truth depth reflects the projection of the underlying 3D object motion. Meanwhile, PSNR, SSIM, and LPIPS measure visual similarity and rollout consistency in addition to rendering quality. Importantly, all these metrics are computed over all rollout frames and averaged across the sequence, so they evaluate simulation quality over time rather than only per-frame reconstruction.
>
> To further assess physical fidelity beyond visual metrics, we additionally report trajectory error (MSE between predicted and reference trajectories) as an explicit motion-based metric. The reference trajectories are estimated using CoTracker3 [1], which serves as a practical proxy for real object dynamics. As shown in Tab. 10, SoMA achieves the lowest trajectory error at all rollout steps, and the advantage remains consistent from short to long horizons, indicating that the simulator captures more accurate object dynamics rather than only visual motion.
>
> ### W3 & Q5 Broader Baselines / Broader representations
>
> A more detailed comparison is summarized in Tab. 8 (In Ans of BTJY). Beyond PhysTwin and GausSim, we additionally consider broader baselines with different 3D representations, including GS-Dynamics and 3DGSim (Gaussian Splats), VPD/HD-VPD (latent space) and PointWorld (point), and DeformContact (mesh). This broader comparison helps clarify both the empirical claims and the role of Gaussian splats in our setting: compared with point/particle/mesh representations, GS provides a better trade-off between simulation and high-fidelity rendering for real-to-sim soft-body manipulation.
> We additionally provide expanded comparisons in Fig.5 (a); since DeformContact does not support rendering, we bind Gaussian splats to mesh points for visualization, denoted as Deform*.
>
> ### W6 & Q2 & Q3: Efficiency Comparison & Scalability
>
> Efficiency is reported in Appendix B.2 (Lines 642–644), including training and inference time, the average inference memory is 2530 MB. Here we additionally provide the inference-time comparison below, since it is the most directly comparable efficiency metric across methods for simulation and planning; all compared methods run at 10–20 FPS, with SoMA at 12 FPS. This shows that SoMA maintains comparable real-time performance while additionally supporting unified real-to-sim mapping, force-driven interaction dynamics, and high-fidelity long-horizon simulation.
>
> **Table 9. Inference-time comparison (FPS) across methods.**
>
> | Inference time | SoMA (ours) | PhysTwin | GausSim | GS-Dynamics | DeformContact |
> |---|---:|---:|---:|---:|---:|
> | FPS | 12 | 10 | 12 | 14 | 20 |
>
> Regarding scalability, SoMA is a neural simulator and therefore scales in the standard data-driven manner to larger datasets and longer trajectories. The main trade-off is increased training and rollout cost as task complexity grows, especially for longer horizons and more complex interactions. In return, the model can support richer manipulation scenarios and long-horizon simulation beyond short-horizon resimulation. Our current experiments already include 100+ frame rollouts and more challenging large-deformation tasks, which provide an initial demonstration of this scalability.
>
> [1] Karaev et al., CoTracker3, ICCV 2025.

---

> > ### Author Rebuttal · Reviewer_GWXV · 2026-04-06
> >
> > I thank the authors for their rebuttal, which has addressed all points raised in my questions.

---

### Official Review · Reviewer_gU9y · 2026-03-10

**Soundness:** 3
**Presentation:** 3
**Significance:** 3
**Originality:** 3
**Overall Recommendation:** 4
**Confidence:** 2

**Summary:**

This paper proposes SoMA, a 3D Gaussian splat-based real-to-sim neural simulator for robotic soft-body manipulation, which achieves deep coupling between robotic manipulation and deformable objects, mitigates the issues of occlusion and instability in existing methods, and offers a novel approach to simulation.

**Compliance With Llm Reviewing Policy:**

Affirmed.

**Final Justification:**

The authors have adequately addressed my questions in their rebuttal, but I believe the generalization ability of the experiments cannot be verified over a broader scope. Therefore, I have maintained my original score.

**Key Questions For Authors:**

1. The contact state between the robot and the deformable object is oversimplified, with the forces exerted by the end-effector and gripper reduced to action points. However, differences in actual gripper models may lead to variations in occlusion and force application. Is the solution proposed in the paper implicitly bound to the specific gripper used, resulting in degraded performance when generalized to human hands or other manipulators?
2. The objects and scenes used in the experiments feature relatively simple visual characteristics. Can the solution in the paper still achieve favorable performance when dealing with deformable objects with patterns or textures?
3. If the same type of object is made of different materials. For example, ropes with an identical visual appearance but different bending stiffness and elastic modulus, will this cause the proposed method to fail?
4. Can the real-time performance of other methods be analyzed in comparison with that of SoMA?

**Limitations:**

Yes.

**Strengths And Weaknesses:**

Strengths:
1. This paper presents a novel R2S simulation approach for the field of robot manipulation, which deeply couples robot joint actions with object and environment dynamics and mitigates the critical occlusion issue to a certain extent.
2. The paper is clearly written, with figures and tables presented in a lucid manner that facilitates easy comprehension.

Weaknesses:
1. The objects and scenes demonstrated in the paper are rather limited in diversity, and my primary concern lies with the generalizability of the method, including that to variations in the visual attributes of objects, object materials, different manipulators and other aspects.
2. The real-time performance of SoMA appears to be inadequate, and we request the authors to further discuss and analyze the trade-offs involved and the associated performance implications.

---

> ### Author Rebuttal · Authors · 2026-03-31
>
> Anonymous Link (Fig.5): https://github.com/anonymousrebuttal2000/anonymous_figure
>
> ### W1 & Q2 & Q3 Generalizability
>
> The generalization ability is shown in Fig. 4(c) with different manipulators and objects, and further supported by the additional experiment in [Fig. 5(b)](https://github.com/anonymousrebuttal2000/anonymous_figure) . We train the model on a small blue cloth, and then estimate the physical attributes using only the first 10 frames while keeping the model parameters fixed. The model generalizes well to a larger gray soft cloth with striped patterns and a larger red stiff cloth with floral patterns, showing robustness to variations in texture, pattern, size, and manipulator.
>
> Our model can handle objects with identical appearance but different materials. In our setting, appearance and material properties are not assumed to be aligned: appearance is reconstructed from the initial frames, while material properties are modeled as learnable per-point latent attributes inferred from the video dynamics.
>
> ### W2 & Q4 Efficiency Performance
>
> The efficiency and real-time performance are discussed in Appendix B.2 (Lines 642–644). We additionally provide the inference-time comparison below; all compared methods run at 10–20 FPS. Compared with prior methods, SoMA maintains comparable real-time performance while additionally supporting unified real-to-sim mapping, force-driven interaction dynamics, and high-fidelity long-horizon simulation, which introduces extra computation but improves controllability and simulation quality.
>
> **Table 9. Inference-time comparison (FPS) across methods.**
>
> | Inference time | SoMA (ours) | PhysTwin | GausSim | GS-Dynamics | DeformContact |
> |---|---:|---:|---:|---:|---:|
> | FPS | 12 | 10 | 12 | 14 | 20 |
>
> ### Q1 Contact mechanism
>
> In our current setting, this contact design is sufficient for robot soft-body manipulation and is not implicitly bound to a specific gripper. We show transfer to other manipulators, including human hands, in Fig. 4(c) and [Fig. 5(b)](https://github.com/anonymousrebuttal2000/anonymous_figure), with stable performance. For more diverse robots or more delicate contact patterns, the end-effector representation can be extended from sparse action points to sampled surface points, allowing the interaction module to better capture gripper geometry and further improve generalization.

---

> > ### Author Rebuttal · Reviewer_gU9y · 2026-04-02
> >
> > I have reviewed the additional results in the appendix. In my opinion, the authors have addressed all the raised questions. However, the generalization to multiple contact points and different gripper devices may require further investigation and additional validation in future work.

---

### Official Review · Reviewer_sQPL · 2026-03-11

**Soundness:** 3
**Presentation:** 2
**Significance:** 3
**Originality:** 3
**Overall Recommendation:** 5
**Confidence:** 4

**Summary:**

Authors propose a neural network model pipeline called "SoMA" that simulates 3D Gaussian splats over time conditioned on a robot's pose. First, standard 3D-GS is used to construct a 3D Gaussian point cloud of a scene at the initial timestep from multi-view camera images. Then graph neural networks predict the change of Gaussian particle parameters using a hierarchical decomposition of the point cloud. Importantly, the model is conditioned on "robot control points" to enable predictions of the scene dynamics conditioned on the robot's state. The model is trained on an image-reconstruction loss using object-centric masking and a hierarchical regularization term that ensures that particles remain close to their "parent" particles.

Simple yet effective experiments demonstrate the aptitude of the model compared to SOTA models.

**Compliance With Llm Reviewing Policy:**

Affirmed.

**Final Justification:**

The authors provided a detailed comparison of the SOTA showcasing the merits of the proposed method. I also appreciate that the authors transparently communicated that the model relies on prior knowledge of the gravity vector and object masks. While these are limitations that must be resolved by future work, I do consider the work an interesting and relevant contribution to the field and recommend an accept.

Minor comments:

    W3: Divide $m(x^{t+1}-x^t)$ by the time step and you have a forward Euler approximation of a particle's momentum. I appreciate that the authors rewrite the paper to avoid claiming the use of a "momentum conservation law".
    How many levels of hierarchy do you need to pass a message in one time step across the full mesh (this is of course a function of the number of message parsing steps)? This question might be interesting, when considering extending this work to rigid body physics.

**Key Questions For Authors:**

- Q1: The authors do not discuss how constructing a point cloud at the first time step affects the practicality of the method. If an object lies on the table, how do you obtain splats for its occluded surfaces?
- Q2: For rigid objects, contact forces propagate instantaneously through the full point cloud. How do you want to model the change in velocity using graph-based point to point message parsing?
- Q3: What is the computational cost of the proposed framework? Could you please add a figure comparing the computational cost of SoMA to GausSim and PhysTwin?
- Q4: How does the masking exactly work? What would happen if you do not use masking during training?
- Q5: Adding the tables normal to the model appears to be strong prior knowledge. Can you remove this information from the model, if yes what would happen?

**Limitations:**

No. See questions above.

**Strengths And Weaknesses:**

**Strengths**
- Excellent experimental results on real-world robot data that compare to recent SOTA models.
- Interesting novel approach to introducing hierarchical structure into a point cloud.

**Weaknesses**

*C1: Work is denotes only a small incremental advance over the state of the art.*
**The work is essentially a combination of PhysTwin /  GaussSim and VPD / HD-VPD**. PhysTwin /  GaussSim resort to 3D-GS to reconstruct 3D Gaussian point clouds which then are simulated through time using an analytical physics solver. VPD proposed to reconstruct point clouds (though not splats) from RGB-D multi-view images and learn particle simulations using GNNs. HD-VPD proposed to leverage an hierarchical structure in the point cloud to improve prediction quality. Unfortunately, these works are not dicussed in the related work, see also C2.

*C2: Related work is not discussing recent advances in neural particle simulation.*
Huang et al ("Point world", Jan 2026) is briefly mentioned under world models, though its also simulating 3D point clouds (though not splats) using a PTv3 point transformer (a transformer can be seen as a special type of graph network). Similarly, Zhobro et al ("3DGSim", Mar 2025) reconstructs 3D-GS particle clouds from multi-view images and simulates them through a PTv3 point transformer.  While both works are "just" arxiv papers, considering how closely these works relate to the author's work, it would benefit the reader if the authors emphasize the differences between SoMA to Point world / 3DGSim. In addition, the authors do not discuss prior work on neural particle simulation. This includes Whitney et al. ("VPD", 2023 and "HD-VPD", 2024). VPD was the first work to learn to simulate 3D particles via neural networks from/on multi-view images. HD-VPD introduces the idea of hierarchical levels of particles which the author also emphasize as main contribution. After VPD, many more works explored the idea of learning to simulate 3D particles such as "Physics-Encoded Graph Neural Networks for Deformation Prediction under Contact" by Saleh. This last work - like many others - also models changes in particle through "forces" predicted by neural networks.

*C3: Wrong terminology?*
I doubt that the authors use of "force" and "momentum" are correct.

It is not clear how Eq. 13 is related to momentum. x after all is the position, while a particle's momentum is mass times velocity. If the authors want to draw a connection between eq. 13 to "momentum", then please make this connection explicit. That said, forcing child particle positions to remain close to their parent's position seems sensible.

In addition, the authors point out at several points in the paper that the network is predicting "forces". However, if I'm not mistaken the network predicts changes in particle velocities. Clearly, following standard Newtonian mechanics, the acceleration of a point's mass arises as the sum of forces divided by the points mass. However, this is not what the network is predicting.

*C4: Unclear model architecture*
I found it difficult to understand how the model is exactly setup. I recommend adding an additional image showing the model architecture. Also I found the paper to be somewhat suboptimally structured. Why is "Section 3: Preliminaries" not part of the model? How do the different part discusses in each sub-section combine to form the whole framework? **I strongly, recommend to rewrite the method section.**

Minor comments:
- "Following prior hierarchical GS simulators" - citations missing.
- Eq. 8 defines g as the "gravity direction" but then refers to it in Eq. 10 as "force"
- Typos: "Defination"
- Undefined symbols: $s_i$, $\mathbf{F}_k^i$, $\mathbf{X}$
- Title of figure 3 uses "comic sans" font?

---

> ### Author Rebuttal · Authors · 2026-03-31
>
> ### W1 & W2  Details comparison of existing methods
>
> A more detailed comparison is summarized in Tab. 8 (In Ans of BTJY). Among closely related methods, PhysTwin and GS-Dynamics are the most relevant to robot-conditioned interaction, while GausSim and 3DGSim focus more on GS-based visual simulation from multi-view observations without explicit 3D tracking supervision. VPD/HD-VPD, PointWorld, and DeformContact instead focus on geometry-driven dynamics with different 3D representations, including latent particles, point clouds, and meshes. Among them, PointWorld and DeformContact do not provide rendering results, while VPD/HD-VPD provides rendered outputs but not at the fidelity of GS-based methods. We additionally provide expanded comparisons in [Fig.5 (a)](https://github.com/anonymousrebuttal2000/anonymous_figure); since DeformContact does not support rendering, we bind Gaussian splats to mesh points for visualization, denoted as Deform*.
>
> ### W1 & W4 Novelty and Writting
>
> Our goal is not simply to combine prior components, but to address a distinct and challenging setting: robot-conditioned real-to-sim neural simulation of complex soft-body manipulation, which is not fully covered by prior methods in terms of both controllable interaction modeling and high-fidelity rendering.
>
> Section 3 introduces the hierarchical Gaussian-Splatting simulation backbone inherited from prior work, including the hierarchical representation and propagation mechanism. Section 4 then presents the SoMA-specific extensions built on top of it. The backbone is a hierarchical GNN that takes current GS states and robot actions as input and predicts next GS states under multi-view supervision and hierarchical physics regularization.
>
> Specifically, Section 4 introduces three key components: (1) R2S mapping, which aligns robot kinematics, object states, and reference frames into a unified simulation space; (2) force-driven dynamics for complex and changing interactions; and (3) the training/supervision design, which improves long-horizon stability, efficiency, and physics-consistent learning. Compared with prior work, which mainly studies simpler motions or less realistic settings, SoMA targets more challenging and application-relevant scenarios such as rope stretching and T-shirt folding. We will revise Section 4 to make this hierarchical composition more explicit and add a dedicated architecture figure illustrating the full pipeline and module interactions.
>
> ### W3 Concerns physics concept detail
>
> We clarify the physical interpretation and terminology used in the paper. Eq. (13) is a physics-inspired hierarchical consistency regularization rather than a strict momentum conservation law; under a fixed discrete-time step, $m(x^{t+1}-x^t)$ can be viewed as a mass-weighted velocity-like quantity.
> For the “force” terminology, the model does not directly predict classical Newtonian forces. Instead, Eqs. (10–11) produce learned interaction signals, which Eq. (9) uses together with the current state to predict next-step velocity and rotation updates. Thus, Eq. (9) is better understood as a discrete-time state update model inspired by Newtonian structure, not a direct physical simulator of forces and accelerations.
>
> ### Minor comments
>
> Thank you for the careful reading. We will revise these minor issues accordingly
>
> ### Q1 Reconstruction Details
>
> The reconstruction details are provided in Appendix B.1 (Lines 623–628). For surfaces occluded by the table, we approximate the unseen bottom part as a planar support surface attached to the table and complete the missing splats with assigned colors.
>
> ### Q2 Global forces propagation
>
> This is exactly why we use a hierarchical graph simulator. At the top level, the graph is fully connected, allowing contact effects to propagate quickly across global clusters through message passing. The resulting responses are then propagated to lower levels and finally distributed to all points, enabling the corresponding velocity changes to spread efficiently through the object.
>
> ### Q3 Computational cost comparison
>
> The computational cost is discussed in Appendix B.2 (Lines 642–644). We additionally provide the inference-time comparison in Tab.9 (In Ans of gU9y); all compared methods run at 10–20 FPS.
>
> ### Q4 The masking mechanism
>
> The masking mechanism is described in Lines 226–236. It excludes occluded regions from image supervision and regularizes them with physics-inspired constraints. As shown in Table 3, the *Img-Only* setting removes masking and physics regularization and uses only full-image loss, leading to the largest degradation in both resimulation and generalization. This shows that masking is important for robust training under partial observability.
>
> ### Q5 Table normal
>
> The main role of the table normal is to align all sequences to a consistent gravity direction during preprocessing. Without this alignment, different samples may have different gravity directions, which hurts generalization

---

> > ### Author Rebuttal · Reviewer_sQPL · 2026-04-02
> >
> > The authors provided a detailed comparison of the SOTA showcasing the merits of the proposed method. I also appreciate that the authors transparently communicated that the model relies on prior knowledge of the gravity vector and object masks. While these are limitations that must be resolved by future work, I do consider the work an interesting and relevant contribution to the field and recommend an accept.
> >
> > Minor comments:
> > - W3: Divide $m(x^{t+1}-x^t)$ by the time step and you have a forward Euler approximation of a particle's momentum. I appreciate that the authors rewrite the paper to avoid claiming the use of a "momentum conservation law".
> > - How many levels of hierarchy do you need to pass a message in one time step across the full mesh (this is of course a function of the number of message parsing steps)? This question might be interesting, when considering extending this work to rigid body physics.

---

> > > ### Author Response · Authors · 2026-04-04
> > >
> > > Thank you for the thoughtful follow-up and for your encouraging comments on our work. We also appreciate your suggestions on readability and will further clarify the presentation.
> > >
> > > Regarding the hierarchical propagation, our current implementation uses 3 hierarchy levels, which provide an efficient way for interaction signals to propagate globally.

---

### Official Review · Reviewer_BTJY · 2026-03-12

**Soundness:** 3
**Presentation:** 3
**Significance:** 3
**Originality:** 3
**Overall Recommendation:** 5
**Confidence:** 2

**Summary:**

SoMA proposes an action-conditioned real-to-sim neural simulator for robotic soft-body manipulation by representing objects as hierarchical 3D Gaussian splats and rolling out dynamics via explicit force-driven updates coupled to robot actions. Across resimulation and generalization to unseen manipulations, it reports consistent improvements over prior baselines, supported by ablations on multi-resolution training and blended supervision.

**Compliance With Llm Reviewing Policy:**

Affirmed.

**Final Justification:**

The author's experiments and visualizations have filled in a lot of gaps; I would like to raise my score.

**Key Questions For Authors:**

1. Could the authors clarify the physical meaning of Eq. (13)? There seems to be a gap between the current formulation and the description of momentum consistency regularization / momentum conservation.

2. Could the authors explain more concretely why GS-Dynamics or other closely related Gaussian-splatting-based neural simulators are not included in the main comparisons?

3. Could the authors provide more direct evidence to isolate the contribution of the force-driven object dynamics relative to a more general state-based formulation?

**Limitations:**

yes

**Strengths And Weaknesses:**

strengths
1. Clear problem setup with practical value. The paper targets an action-conditioned neural simulator for real-to-sim soft-body manipulation, rather than generic dynamic reconstruction. It unifies robot-conditioned real-to-sim mapping, soft-body dynamics, and robotic manipulation within a single framework and setting.
2. Coherent and well-structured method. The architecture is not a loose collection of modules. The main text and Fig. 2 clearly delineate the roles of the robot-conditioned R2S mapping, force-driven updates defined directly on Gaussian splats, the multi-resolution training strategy, and the blended supervision scheme, forming a consistent overall pipeline.

weaknesses
1. The physical interpretation of Eq. (13) is confusing. The equation is written as $m\hat{x}$, while the text describes it as a momentum consistency regularization and even momentum conservation. Under the paper’s notation, it appears closer to a hierarchical mass-weighted position consistency term rather than true momentum conservation, which weakens the “physics-inspired consistency constraints” narrative.

2. Positioning w.r.t. closest related methods is not sufficiently clear. The paper groups GausSim and GS-Dynamics under “neural simulators on GS representations,” yet the main experiments compare only against PhysTwin and GausSim. The baseline justification remains largely at the category level and does not sufficiently explain how these neighboring methods differ from the present setting in terms of robot-conditioned interaction, control-aware dynamics, interaction modeling, or scene-to-simulation mapping.

3. The advantage of force-driven dynamics is supported mostly indirectly. Current evidence is primarily system-level outcome improvements, with limited module-level validation. Comparisons also focus mainly on PhysTwin and GausSim, and the evaluation is still limited in scale. As a result, the results support “effective under the current setup” more than stronger claims of general applicability.

---

> ### Author Rebuttal · Authors · 2026-03-31
>
> Anonymous Link (Fig.5): https://github.com/anonymousrebuttal2000/anonymous_figure
>
> **Table 8. Broader Baseline**
> |                      | SoMA (ours)      | PhysTwin                  | GS-Dynamics                 | GausSim         | 3DGSim [1]         | VPD/HD-VPD [2,3]                   | PointWorld  | DeformContact [4] |
> |----------------------|------------------|---------------------------|-----------------------------|-----------------|-----------------|-------------------------------|--------------|-----------------|
> | Real or Virtual Data | **Real**         | **Real**                  | **Real**                    | **Real**        | Virtual         | **Real**                      | **Both**     | Virtual       |
> | Dynamic Pattern      | **Robot Manipulation** | Hand Manipulation         | **Robot Manipulation**      | Deformation     | Collision & Deformation | **Robot Manipulation**       | **Robot Manipulation** | Deformation   |
> | Action-conditioned   | **yes**          | **yes**                   | **yes**                     | no              | no              | **yes**                       | **yes**      | no            |
> | 3D Representation    | Gaussian Splats  | Gaussian Splats & Point Cloud | Gaussian Splats & Control Points | Gaussian Splats | Gaussian Splats | Latent Particles & Point Cloud | Point Cloud  | Mesh          |
> | Training Supervision | Multi-view Videos | Multi-view Videos & 3D tracking | Multi-view Videos & 3D tracking | Multi-view Videos | Multi-view Videos | Multi-view Videos            | 3D tracking  | /             |
> | Rendering Results    | **yes**          | **yes**                   | **yes**                     | **yes**         | **yes**         | Coarse                        | no           | no            |
> | Open Source          |                  | **yes**                   | **yes**                     | **yes**         | no              | no                            | part (no ckpt) | **yes**      |
>
> ### W1 & Q1 Concerns about "physics-inspired consistency constraints"
>
> We would like to clarify the physical interpretation of Eq. (13). In discrete time, $m(x^{t+1}-x^t)/\Delta t$ is closely related to momentum. Eq. (13) can thus be interpreted as promoting consistency in such mass-weighted motion patterns, rather than explicitly modeling momentum conservation. We intend to introduce a physics-inspired regularization that encourages temporally coherent motion across the hierarchy. We acknowledge that the current wording may overemphasize the connection to momentum conservation, and we will revise the text to better reflect this interpretation.
>
> ### W2 & Q2 Lack details comparison of baselines
>
> We clarify our baseline selection and the positioning of closely related methods such as GS-Dynamics (above table). Among closely related methods, PhysTwin and GS-Dynamics are the most relevant to robot-conditioned interaction, while GausSim and 3DGSim focus more on GS-based visual simulation from multi-view observations without explicit 3D tracking supervision, thus avoiding errors and bias from intermediate tracking. VPD/HD-VPD, PointWorld, and DeformContact instead focus on geometry-driven dynamics with different 3D representations, including latent particles, point clouds, and meshes. Among them, PointWorld and DeformContact do not provide rendering results, while VPD/HD-VPD provides rendered outputs but not at the fidelity of GS-based methods. Compared with prior GS-based neural simulators, our main advance is extending the setting from simpler motions to complex robot-object interactions with large deformations, such as T-shirt folding (Fig. 1). To make the comparison more complete, we additionally provide results for other open-sourced methods in Fig. 5(a). Since DeformContact does not support rendering, we bind Gaussian splats to mesh points for visualization, denoted as Deform*.
>
> ### W3 & Q3 Force-driven / Limited span
>
> *Force-driven*: Fig. 3 provides a controlled comparison between a purely state-based formulation (e.g., GausSim-style) and our force-driven dynamics, where the key difference is the inclusion of action-conditioned interaction signals. The results show that state-based dynamics can handle simple near-constant motions (e.g., cloth lifting), but struggles with temporally varying interactions and complex deformations (e.g., rope stretching), where explicit force conditioning becomes critical. We note that other components (e.g., multi-resolution training and physics supervision) mainly contribute to long-horizon stability and consistency, while the primary gain in interaction modeling comes from the force-driven formulation.
>
> *More Evaluation*: Fig. 5(a) shows more baselines, while Fig. 4(c) and Fig. 5(a) show generalizability across different objects, materials, patterns, and manipulators.
>
> [1] Zhobro et al., 3DGSim, arXiv 2025.
>
> [2] Whitney et al., VPD, arXiv 2023.
>
> [3] Whitney et al., HD-VPD, CoRL 2025.
>
> [4] Saleh et al., DeformContact, ICRA 2024.

---

> > ### Author Rebuttal · Reviewer_BTJY · 2026-04-05
> >
> > Fully resolved

---

### Decision · Program_Chairs · 2026-04-30

**Decision:**

Accept (regular)

**Comment:**

This paper received a unanimously positive recommendation from all reviewers after the post-rebuttal discussion. The reviewers initially raised concerns with the paper's technical exposition, its difference from related work, and its experimental evidence. The rebuttal has fully resolved all reviewers' major concerns with the paper. Therefore, I recommend that ICML accept this paper.